DOI: 10.1038/s41467-017-01406-6　　OPEN

# Anaerobic microsites have an unaccounted role in soil carbon stabilization

Marco Keiluweit[1,2], Tom Wanzek[3], Markus Kleber[3,4], Peter Nico[5] & Scott Fendorf [2]

Soils represent the largest carbon reservoir within terrestrial ecosystems. The mechanisms controlling the amount of carbon stored and its feedback to the climate system, however, remain poorly resolved. Global carbon models assume that carbon cycling in upland soils is entirely driven by aerobic respiration; the impact of anaerobic microsites prevalent even within well-drained soils is missed within this conception. Here, we show that anaerobic microsites are important regulators of soil carbon persistence, shifting microbial metabolism to less efficient anaerobic respiration, and selectively protecting otherwise bioavailable, reduced organic compounds such as lipids and waxes from decomposition. Further, shifting from anaerobic to aerobic conditions leads to a 10-fold increase in volume-specific mineralization rate, illustrating the sensitivity of anaerobically protected carbon to disturbance. The vulnerability of anaerobically protected carbon to future climate or land use change thus constitutes a yet unrecognized soil carbon–climate feedback that should be incorporated into terrestrial ecosystem models.

[1] School of Earth and Sustainability and Stockbridge School of Agriculture, University of Massachusetts, 411 Paige Lab, Amherst, MA 01003, USA. [2] Earth System Science Department, Stanford University, Via Ortega 473, Stanford, CA 94305, USA. [3] Department of Crop and Soil Science, Oregon State University, 317 ALS Building, Corvallis, OR 97331, USA. [4] Institut für Bodenlandschaftsforschung, Leibnitz-Zentrum für Agrarlandschaftsforschung (ZALF) e. V., Eberswalder Straße 84, 15374 Müncheberg, Germany. [5] Earth and Environmental Sciences Area, Lawrence Berkeley National Laboratory, Building 85B, Berkeley, CA 94720, USA. Correspondence and requests for materials should be addressed to S.F. (email: fendorf@stanford.edu)

Despite the important role of the vast soil carbon stocks in the global carbon cycle[1–3], disagreement persists regarding the effects of climate and land use change[2,3]. This is an urgent debate because positive feedbacks to climate change are projected to occur if soil carbon is released to the atmosphere as $CO_2$ through enhanced microbial respiration (or mineralization)[4]. While attempts have been made to predict the timing and severity of soil carbon–climate feedbacks[5], a potentially important feedback mechanism has heretofore been unrecognized: the release of carbon currently stored within oxygen-deprived, anaerobic soil microsites in generally well-aerated soils.

The prevailing paradigm assumes that upland mineral soils (denoting soils not falling within the hydric definition) are well aerated. Hence, aerobic heterotrophic respiration, which is primarily responsible for $CO_2$ emissions from soils, is the exclusive metabolism. Consequently, contemporary carbon-cycling models thus far[6] are deemed to be insignificant[7] or insufficiently account for[8,9] the potential impact of anaerobic microsites on mineralization rates. If oxygen diffusion into structural units such as peds or aggregates is slower than its consumption by microbes, oxygen is depleted, and anaerobic microsites are established[10], prompting a switch from aerobic respiration to a diversity of anaerobic pathways[11]. In permanently anaerobic environments such as marine sediments[12], wetlands, or peatlands[13], mineralization rates dramatically decline compared to aerobic environments, frequently by 60–95%[14]. This limitation is often attributed to the inhibition of oxidative enzymes involved in depolymerization of macromolecules[15], or lower energy yields associated with alternate electron acceptors (e.g., Fe(III) or S(IV)) in microbial respiration[16]. Mineralization rates may further decline with oxidation of more reduced organic substrates when coupled with alternate electron acceptors[17]. The question remains, however, whether anaerobic microsites within soils could impart the selective preservation of reduced compounds such as lipids and waxes in upland soils globally[18–21].

Here, we show that anaerobic microsites in otherwise well-aerated soils impose drastic metabolic constraints on mineralization rates, and impart long-term protection to reduced organic compounds. Using artificial and natural gradients in particle size distribution, we examine the effect of anaerobic microsites on microbial metabolism and mineralization rates through well-controlled laboratory experiments, and we assess their impact on the persistence of organic compounds within a representative upland soil ecosystem. Further, we use derived anaerobic and aerobic mineralization rates to project the vulnerability of soil carbon stocks in anaerobic microsites. Both experimental and field systems show remarkably consistent trends, highlighting that anaerobic microsites effectively protect soil carbon against microbial decomposition in upland soils.

## Results

**Investigating anaerobic microsite effects on metabolic rates**. To isolate the contribution of anaerobic microsites on soil carbon cycling, and to determine volume-averaged mineralization rates, we conducted experiments within flow reactors where variations in particle size distribution were used to systematically alter oxygen supply. The reactors consisted of an advective flow domain receiving a continuous supply of oxygen-saturated soil solution in contact with a diffusion-limited domain packed with soil (Supplementary Fig. 1). Soil was mixed (1:1) with quartz ground to two different particle sizes (25–45 and 150–250 μm); finer particle sizes decreased the effective diffusion length into the soil, and thus increased the extent of anaerobic microsites relative to coarser particles (Fig. 1a).

Carbon mineralization rates, measured as $CO_2$ production in the effluent, were inversely proportional to the extent of the anaerobic microsites over the course of the flow experiment (Fig. 1b). Expanding the anaerobic domain by 5% (in volume) decreased the overall mineralization rates by 37% (Fig. 2a). The overall aerobic respiration rates decreased by the same margin (41%, Fig. 2b). In general, volume-specific anaerobic respiration rates were 10-fold lower than aerobic rates (Fig. 2c). Thus, because aerobic respiration operated at much greater volume-specific rates than anaerobic respiration, a relatively small change in oxygen availability has a disproportionate impact on the overall mineralization rates.

**Metabolic constraints on carbon oxidation**. Oxygen limitations induced successive electron acceptor pathways (e.g., nitrate, Mn, and Fe reduction), with dissimilatory Fe(III) reduction remaining as the predominant anaerobic metabolism (>79–82% of anaerobic respiration) (Fig. 2d). By considering thermodynamic yields of

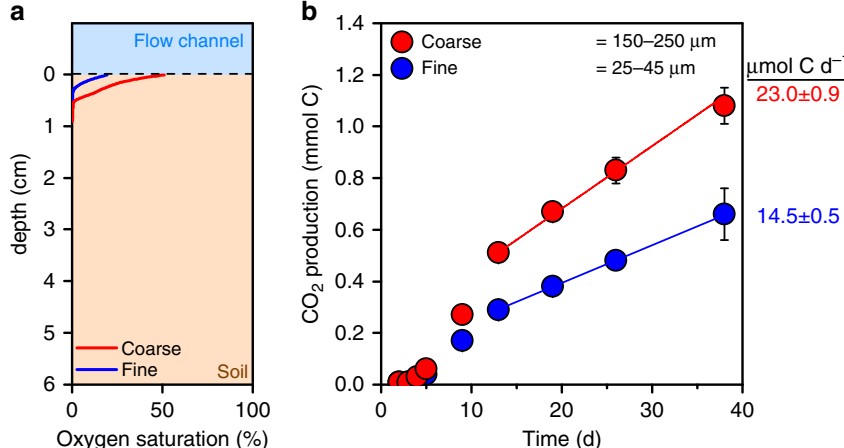

**Fig. 1** Microsite impacts on metabolic rates in flow reactor experiments. **a** Particle size-induced variations in the anaerobic pore volume. Oxygen profiles in reactor experiments conducted with soil amended with quartz grains of different particle sizes (coarse = 150–250 μm and fine = 25–45 μm). Oxygen profiles are the average of three replicate profiles recorded with microelectrodes on day 35 of the incubation. **b** Particle size-induced variations in $CO_2$ production (i.e., mineralization) over the incubation period. Linear fits indicate the incubation period used to calculate mineralization rates. Error bars denote the standard error of the mean calculated from four replicate reactors

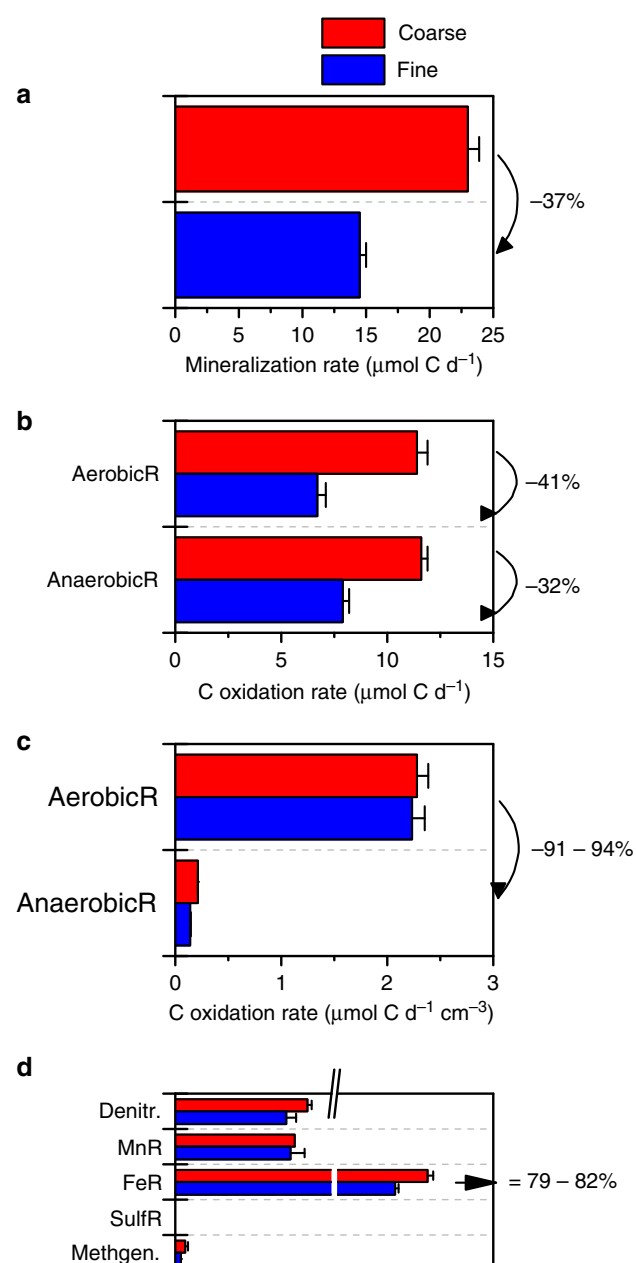

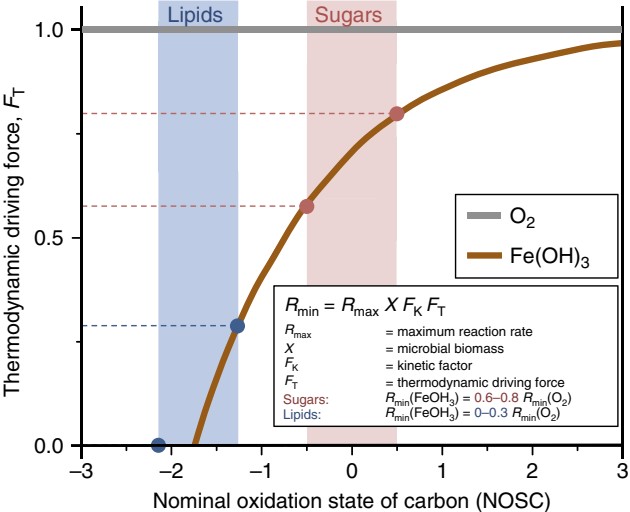

**Fig. 3** Bioenergetic projections for the mineralization of organic compounds under aerobic and anaerobic conditions. The thermodynamic driving force, $F_T$, is given for the oxidation of organic compounds spanning a range of nominal oxidation states when coupled to the reduction of oxygen (aerobic respiration) or $Fe(OH)_3$ (anaerobic respiration). When coupled to oxygen (gray line), $F_T$ for the oxidation of an organic compound is close to 1. Mineralization ($R_{min}$) is thus expected to proceed uninhibited for compounds spanning the full range of oxidation states. Under $Fe(OH)_3$-reducing conditions (orange line), mineralization of more oxidized compounds, such as sugars, yields projected rates comparable to those under aerobic conditions (60–80%). By contrast, mineralization of reduced organic compounds, such as lipids, would proceed at limited rates or may even be completely inhibited (0–30%), resulting in a relative enrichment, and ultimately preservation, of these abundant compound classes in anaerobic microsites

**Fig. 2** The effect of particle size-induced variations in the extent of anaerobic microsites on microbial metabolic rates and pathways. **a** Overall mineralization rate within the reactor, **b** comparison of the total aerobic and anaerobic respiration rates, **c** volume-specific aerobic and anaerobic respiration rates, and **d** contribution of anaerobic respiration pathways to the overall rates. The overall mineralization rates were calculated using linear fits, as shown in Fig. 1a. A combination of anaerobic incubations, solid- and solution-phase measurements, and a mass balance approach was used to quantify aerobic and anaerobic respiration rates across the diffusion-limited domain (a detailed description of our approach is provided in the "Methods" section). Particle size variations to manipulate the extent of anaerobic microsites are shown in red (coarse = 150–250 μm) and blue (fine = 25–45 μm). Error bars denote the standard error of the mean calculated from four replicate reactors

carbon oxidation under Fe-reducing conditions and bioenergetics rate formalism[22], we can project the decline in mineralization rate for different compound classes under anaerobic conditions. As derived by Jin and Bethke[23], C mineralization rate ($R_{min}$) can be expressed as

$$\text{Rate}_{min} = R_{max} X F_K F_T \qquad (1)$$

where $R_{max}$ and $X$ are the maximum reaction rate and microbial biomass, respectively. The functions $F_K$ and $F_T$ are nondimensional and vary between 0 and 1. $F_K$ represents a microbe's ability to acquire and process reactants, thus accounting for enzyme kinetics expressed within the typical Michaelis–Menten or Monod equations; it also encompasses mineral protection and physical isolation. The catabolic energy yield, $F_T$, links the rate of reaction to the thermodynamic driving force. The function $F_T$ represents a fundamental bioenergetic control on microbial oxidation of organic compounds, and is dependent on the Gibbs free-energy term, $\Delta G_{rxn}$, for the overall reaction. $\Delta G_{rxn}$ is dependent on the reduction half-reaction of the terminal electron acceptor, and the oxidation half-reaction for the organic carbon substrate. LaRowe and van Cappellen[17] recently gave a formalism relating $\Delta G_{rxn}$ for the oxidation of a given organic compound to its nominal oxidation state of carbon (NOSC). This concept provides an easy means to calculate the thermodynamic driving force, $F_T$, for the mineralization of an organic compound with a given NOSC when coupled to the predominant electron acceptors in the system (here oxygen or $FeOH_3$) (Fig. 3). When coupled to oxygen, $F_T$ is close to 1, and the reaction is expected to proceed almost uninhibited for compounds spanning the full range of NOSCs. Consequently, the $R_{min}$ would be the same for reduced

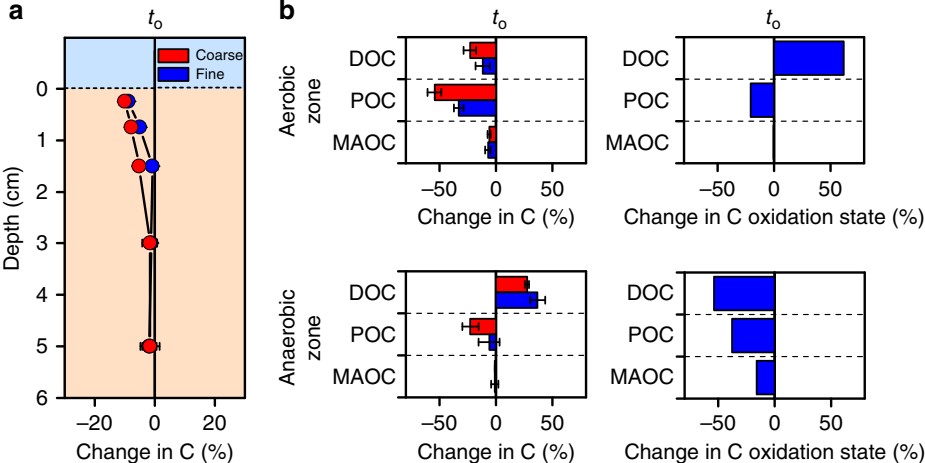

**Fig. 4** The effect of particle size-induced variations in the extent of anaerobic microsites on the microbial oxidation of different carbon pools. **a** Changes in total carbon across the diffusion-limited domain within the reactor. **b** Changes in dissolved, particulate, or mineral-associated organic carbon pools in the aerobic and anaerobic zone. Dissolved (DOC), particulate (POC), and mineral-associated organic carbon (MAOC) are listed in order of decreasing bioavailability. **c** Changes in carbon oxidation state in reactor experiments. The changes in oxidation state are calculated as the ratio of the absorbance of carboxyl (288.35 eV) to that of aromatic C (285.05 eV) in spectra obtained by near-edge X-ray absorption fine structure (NEXAFS) spectroscopy. Changes in carbon pools and oxidation state are shown relative to a time-zero control ($t_0$). Particle size variations to manipulate the extent of anaerobic microsites are shown in red (coarse = 150–250 μm) and blue (25–45 μm). Error bars denote the standard error of the mean calculated from four replicate reactors

compounds with low NOSC (e.g., lipids), and more oxidized compounds with high NOSC (e.g., sugars). By contrast, when oxidation of these compounds is coupled to the reduction of Fe (OH)$_3$, the dominant electron acceptor in anaerobic microsites (Fig. 2d), a very different scenario results. For more oxidized compounds such as sugars with NOSCs of around zero, $F_T$ decreases to 0.6–0.8, reducing Rate$_{min}$ by 20–40% relative to aerobic conditions. This decrease is even more dramatic for more reduced organic compounds such as lipids. $F_T$ for lipids with NOSCs of less than −1.7 is zero, and mineralization ($R_{min}$) is thermodynamically limited if not completely inhibited (Fig. 3). These bioenergetic constraints may result in relative enrichment, and ultimately preservation, of abundant, reduced compound classes such as lipids in anaerobic microsites[24].

Supporting our bioenergetic projections, an increase in anaerobic domain size resulted in greater remaining total carbon content (Fig. 4a). Furthermore, in comparison to mineral-associated carbon, dissolved and particulate organic carbon were mineralized rapidly in the aerobic zone (Fig. 4b), but remained relatively unchanged in the anaerobic zone (Fig. 4b), suggesting that anaerobic microsites efficiently protect otherwise inherently bioavailable carbon pools. Carbon compounds also had a higher oxidation state in the aerobic than the anaerobic zone (Fig. 4c), revealing the selective preservation of reduced carbon compounds under anaerobic conditions. Moreover, the oxidation state of dissolved and particulate organic carbon noticeably declined in the anaerobic zone (Fig. 4c). Our results therefore indicate that thermodynamic constraints on microbial metabolism selectively preserve otherwise bioavailable, reduced compounds in anaerobic microsites, offering a distinct chemical signature indicative of anaerobic protection of soil carbon.

**Anaerobically protected carbon in upland soils**. The protective effect of anaerobic microsites on soil carbon was further examined in upland soils (Hyslop Experimental Station, Corvallis, OR, USA) differing in particle size distribution (texture). We contrasted a finer-textured with a coarser-textured soil, which was otherwise comparable with respect to climate, vegetation, rates and types of carbon inputs, and mineral composition (see

"Methods" section for details); the fine-textured soil showed significantly greater amounts of water-extractable and particulate organic carbon, and a lower average redox potential than the coarse-textured soil (Fig. 5a and Supplementary Fig. 2). Carbon 1s near-edge X-ray absorption fine structure (NEXAFS) analysis showed that bulk and dissolved carbon in the fine-textured soil have a lower average oxidation state (Fig. 5b and Supplementary Fig. 3), and a significant enrichment in reduced functional groups such as aromatic and aliphatic C relative to coarse-textured soil (Fig. 5c). In support of this finding, $^{13}$C nuclear magnetic resonance (NMR) spectra of water extracts revealed that the relative abundance of alkyl C in the fine-textured subsoil is three times greater than the amount present in coarse-textured subsoil (Supplementary Fig. 4). Alkyl C in $^{13}$C NMR spectra is generally attributed to aliphatic compounds, such as lipids, waxes, and fatty acids[5], suggesting the preferential preservation of these water-extractable (and thus potentially bioavailable), reduced compounds in finer-textured upland soils. Consistent with a greater abundance of reduced compounds, Fourier-transform ion cyclotron resonance mass spectrometry (FT-ICR-MS) analysis of water and methanol extracts showed that the average NOSC of the detected organic compounds in fine-textured soil is lower than that in coarse-textured soil (Fig. 6). These field observations are consistent with the reactor results, and support our contention that anaerobic microsites effectively protect bioavailable, reduced organic compounds against decomposition.

## Discussion

The finding that highly reduced organic compounds accumulate in anaerobic microsites promises to resolve an ongoing debate over the mechanisms controlling the long-term stabilization of carbon in soils. Reduced compounds, such as cutin, suberin, waxes, and lipids, have been identified as the most persistent organic compounds in many soils[18–21]. Such reduced compounds accumulate in finer particle size fractions[18,25], in subsoils[26], and with increasing precipitation[27]—all suggestive of oxygen limitations within microsites. Consistent with our thermodynamic predictions (Fig. 3), bioenergetic constraints lead to the selective protection of reduced carbon compounds, such as waxes and

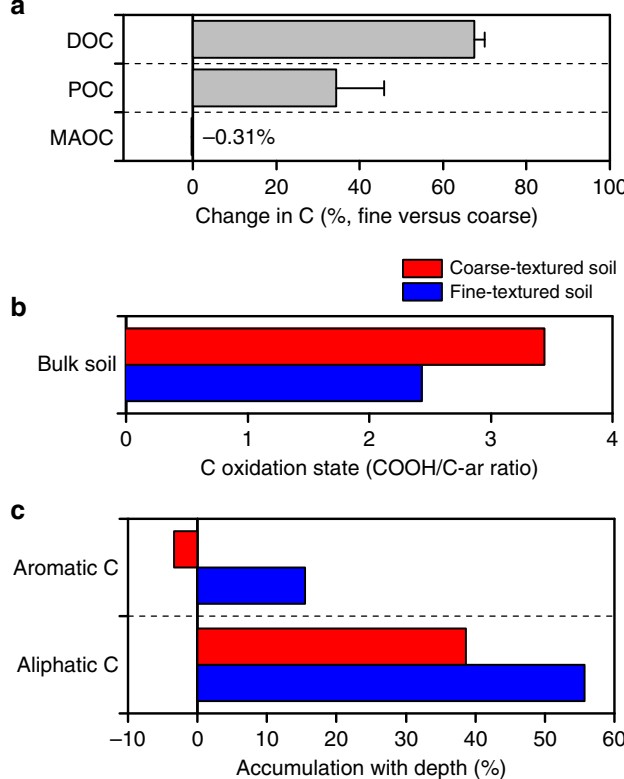

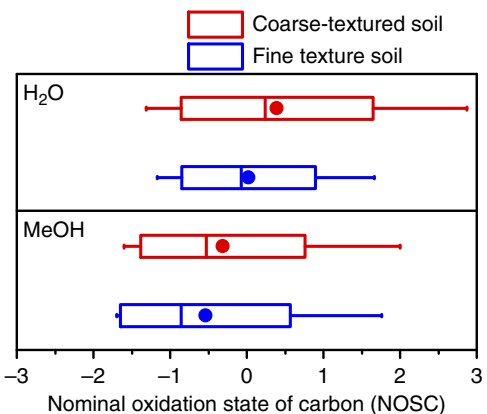

**Fig. 6** Nominal oxidation state of carbon in fine- and coarse-textured upland soil varying in the abundance of anaerobic microsites. The boxes enclose the interquartile range, the whiskers indicate minimum and maximum values, horizontal bars dissecting the boxes represent the median values, and the symbols (dot) represent the mean. Carbon in fine-textured soil showed a lower oxidation state (i.e., it consisted of more reduced carbon compounds) than the coarse-textured soil when extracted with water ($H_2O$) or methanol (MeOH). The average nominal oxidation state of carbon (NOSC) was calculated based on molecular formulae provided for water and methanol extracts analyzed by FT-ICR-MS. Details on the FT-ICR-MS analysis and NOSC calculations can be found in the "Methods" section

**Fig. 5** Impact of soil particle size distribution, and the associated variations in anaerobic soil volume, on carbon bioavailability and chemistry. **a** Differences in the dissolved (DOC), particulate (POC), or mineral-associated organic carbon (MAOC) pools between coarse- and fine-textured soil. DOC, POC, and MOAC are shown as the relative differences between the two soils. **b** Differences in carbon oxidation state between fine- and coarse-textured soil. Carbon oxidation state is calculated as the ratio of the NEXAFS absorbance of carboxyl (288.35 eV) to that of aromatic C (285.05 eV). **c** Differences in the abundance of reduced carbon functional groups between fine- and coarse-textured soil and their accumulation with depth. Changes in the abundance of NEXAFS-detectable aromatic and aliphatic C in the subsoil (100 cm) are shown relative to subsoil horizons (20 cm) (see the "Methods" section for details on NEXAFS analysis). Error bars denote the standard error of the mean calculated from the analysis of three replicate soil cores

lipid, in anaerobic microsites. Our experimental results therefore suggest that oxygen limitations may explain the long residence times of certain organic compounds in upland soils that were previously thought to be the result of recalcitrance. The contribution of bioenergetic constraints relative to, or in combination with, other protection mechanisms—such as chemical associations with mineral surfaces, and the physical isolation of decomposers and substrates—will be dependent on soil and site properties; at present, the contributions and interactions are in need of being resolved.

Current soil carbon models assume compound-specific decomposition rates irrespective of the present environmental conditions[6,7,8,28]. We show that predictions of the vulnerability of specific compounds to disturbance would be much improved by considering oxygen limitations and the underlying thermodynamic controls on decomposition in soil. While recent attempts show promise[29,30], our results suggest that explicit consideration of (i) diffusion-limited domains within the soil structure (both macro- and microaggregates) and (ii) bioenergetic constraints on the mineralization of specific compound classes

(based on the oxidation state of carbon) would further improve model estimates of the impact of anaerobic microsites on decomposition rates.

The dynamics of anaerobic microsites in reasonably well-drained soils are frequently overlooked. Routine or bulk measurements often give the impression that soils are well aerated, although active anaerobic metabolism (e.g., Mn and Fe reduction, or methanogenesis) indicates the presence of anaerobic microsites in upland soils[24]. While quantifying anaerobic microsites remains at the nascent stage, the strong response of mineralization rates to subtle shifts in the extent of anaerobic pore volume illustrates the sensitivity of anaerobically protected soil carbon to sudden changes in soil aeration and the importance of resolving it. In our system, a simple switch from anaerobic to aerobic respiration caused a 10-fold increase in volume-specific mineralization rates (Fig. 2c). Consequently, marginally reducing the anaerobic pore volume in upland soils could result in a disproportionate increase in mineralization rates. These considerations suggest that carbon stocks in upland soils may be vulnerable to disturbances that enhance soil aeration and release previously protected soil carbon as $CO_2$. To assess the potential magnitude of this effect, quantitative data documenting the abundance of anaerobic microsites and the amount of carbon protected therein are urgently needed.

Current conceptions and numerical models assume that long-term carbon storage in upland soils is primarily due to chemical associations of organic compounds with minerals or the physical separation of organic compounds from microbes[3,31], mechanisms relatively insensitive to changing soil temperature[32] or moisture regimes[33], and are far less sensitive to disturbance than anaerobic microsites. Soil aeration can be dramatically impacted by climate events (e.g., drought) and change (increasing temperature). Further, land use change, such as invoking tillage or other physical disturbances, increases soil aeration, diminishing anaerobic carbon protection within microsites. If other protection mechanisms compensate for these disturbance effects, or if anaerobically protected C pool is lost as $CO_2$, it poses an intriguing research

question for the future. Our results demonstrate, however, that without recognizing the importance of anaerobic microsites in protecting soil carbon in soils, we massively underestimate the vulnerability of this vast carbon reservoir to disturbance induced by climate or land use change.

## Methods

**Soil characteristics.** To isolate the impacts of anaerobic microsites on carbon dynamics, we selected a site with variations in particle size distribution (or texture), while having other environmental/soil factors (e.g., C input, pH, temperature, and rainfall) constant. Soils in the Willamette Valley, OR, USA, have developed from stratified, silty glaciolacustrine sediments that were deposited during the late Pleistocene. The brief but intense Missoula flood events created a small-scale variability in texture among otherwise very similar parent materials, resulting in a family of soils covering a gradient in texture. At the OSU Hyslop Field Research Lab (latitude/longitude: 44° 38′ 03″/123° 11′ 24″), these gradients are expressed within the proximal Willamette (well drained), Woodburn (moderately drained), and Amity (somewhat poorly drained) soil series. The mean annual precipitation is 1092 mm, and the mean annual temperature is 11.1 °C. During the wet winter months, the drainage gradients cause a clear variation in redox potential (Supplementary Fig. 2). These arable soils are classified as Mollisols (Supplementary Table 1 for a detailed classification of each soil series). Mollisols primarily occur in the middle latitudes and are extensive in prairie regions such as the Great Plains of the United States. Globally, they occupy ~7.0% of the ice-free land area and are the soil order where the majority of the world's grain harvest is produced. In the United States, they are the most extensive soil order, accounting for ~21.5% of the land area. We collected top- (depth = 10–30 cm) as well as subsoil (depth 90–110 cm) material, which was sieved (<2 mm), air-dried, and stored until further use.

The basic soil characteristics of the two upland soils contrasted in this study are shown in Supplementary Table 1. Soil analyses were performed following standard procedures[34]. Soil pH was measured on field-fresh soil samples with a combination electrode in water at a soil:solution ratio of 1:2.5. The suspension was shaken for 5 min and was allowed to equilibrate for 1 h. To measure the particle size distribution, air-dried and sieved (2 mm) soil samples were treated with $H_2O_2$ to remove organic matter and then dispersed with sodium hexametaphosphate. The particle fractions were determined using a modified pipette method. To determine the total carbon and nitrogen, samples were dried (40 °C) and ground with an agate mill. An elemental analyzer (NA1500, Carlo Erba) was used to determine carbon and nitrogen. The cation exchange capacity (CEC) was determined as the sum of exchangeable cations (effective CEC = Σ Ca, Mg, K, Na, and Al) at soil pH. Dried (30 °C) and sieved (2 mm) samples were extracted with 1 M $NH_4Cl$. The elements were determined using ICP-OES.

**Reactor experiments.** Flow experiments were conducted in reactors (dimensions = 100 (W) × 80 (H) × 10 (D) mm) designed and fabricated to deliver a constant supply of oxygenated nutrient solution through the advective flow channel to soils varying only in texture (Supplementary Fig. 1). Packing reactors with equal amounts of soil, and different-sized quartz grains varied the texture artificially. We chose topsoil material from the Woodburn soil series (fine silty-mixed superactive mesic Aquic Argixeroll) for the reactor experiment. We expected this soil to establish detectable metabolic gradients rather quickly, as field measurements had shown the greatest seasonal variations in redox state. Different size fractions of acid-washed quartz sand (25–45, 75–105, and 150–250 μm) were obtained by sieving, which were then mixed with soil at a 1:1 ratio (weight basis). Clay and silt contents of the resulting mixture were ~21 and ~80% greater in the finest mixture than in the coarsest mixture, as determined by a Coulter LS 230 particle size analyzer. The reactors were wet-packed with the soil–quartz mixture, and the mixture was allowed to settle for 2 days prior to the start of the flow experiment.

The inflow was connected to a peristaltic pump, which provided sterile, inorganic nutrient solution at a constant rate. The solution composition was chosen to mimic that of water extracts from the soil (23 μM $CaCl_2$, 8 μM KCl, 10 μM NaCl, and 5 μM $NaH_2PO_4$). The reservoir containing the nutrient solution was continuously bubbled with air to achieve complete saturation with oxygen (~256 μmol $L^{-1}$). A flow rate of 1.75 ml $h^{-1}$ (equivalent to ~0.5 reactor volumes per day) was chosen because it resulted in sufficient oxygen diffusion into the soil without resuspending soil particles. Gas-tight fittings and PEEK tubing were used throughout. Incubations were carried out at 25 °C in the dark for 40 days. Four replicate reactors were assembled for each treatment.

The effluent was collected in airtight amber serum vials. $CO_2$ and $CH_4$ concentrations in the headspace were measured by gas chromatography (Shimadzu GC-2014) and used to calculate DIC concentrations in the effluent based on Henry's law. DOC concentrations in the effluent were measured on a total organic carbon analyzer (Shimadzu TOC-L).

Oxygen, pH, and $H_2S$ concentration profiles were recorded using 100-μm tip microsensors (OX-100, Unisense, Denmark) inserted through specially designed ports on the top of the reactor. Linear calibrations were performed according to the manufacturer's instructions. For the measurement, the rubber stopper sealing the port was removed, and the microsensor, mounted on a micromanipulator (MM33-

2, Unisense) and connected to a picoamperemeter (PA-2000, Unisense), was slowly inserted. Using the complementary software package (Sensortrace, Unisense, Denmark), profiles across the diffusion-limited domain were recorded to a depth of 5 cm. For pH measurements, the reference electrode was inserted into a separate port. Pore water samples were collected using microrhizon samplers (MicroRhizons, Rhizosphere Research Products, the Netherlands) placed at different depths (depth = 0.5, 1, and 4 cm) within the diffusion-limited domain in the reactor. At each depth, ~1 mL was extracted into pre-evacuated amber serum vials. At the end of the flow experiment, reactors were opened in an anaerobic glove box, and six depth increments within the diffusion-limited domain (0–0.5, 0.5–1, 1–2, 2–3, 3–4, and 4–6 cm) were carefully sampled for solid- and solution-phase analysis of alternative terminal electron acceptors, anaerobic incubations, and SOC characterizations.

**Aqueous and solid-phase analysis.** Extractable nitrate and ammonium concentrations at each depth were determined in duplicates using a flow injection analyzer (SmartChem 200 Discrete Analyzer, Westco). Mn(II) and Fe(II) concentrations were determined by duplicate extractions with 0.5 M HCl. HCl-extractable; Mn was assumed to represent $Mn^{2+}$, which was analyzed by ICP-OES (ICAP 6300 Duo View, Thermo Scientific), and $Fe^{2+}$ in the extract was determined using a phenanthroline assay.

**Carbon oxidation rate measurements and calculations.** Electron acceptor profiles (Supplementary Fig. 5) were used in combination with the incubations to calculate the relative contributions of different respiration pathways. Our mass balance approach was modified from Canfield et al.[35,36], and is summarized in Supplementary Table 2.

During the incubation, the effluent was collected in airtight amber serum vials. $CO_2$ and $CH_4$ concentrations in the headspace were measured by gas chromatography (Shimadzu GC-2014) and used to calculate DIC concentrations in the effluent based on Henry's law. DOC concentrations in the effluent were measured on a total organic carbon analyzer (Shimadzu TOC-L). DOC and $CH_4$ profiles in the effluent are shown in Supplementary Fig. 6.

**Anaerobic respiration.** The overall anaerobic respiration rates were estimated by anaerobic incubations of soil taken from different depth increments below the aerobic zone (depth <0.5 cm) upon reactor harvest. In the anaerobic glove bag, 3 g of soil from each depth increment was transferred to 35-mL serum vials, and was thoroughly mixed with 3 mL of anoxic Milli-Q $H_2O$. The vials were capped with butyl rubber stoppers and the vials were incubated in the dark for 48 h. The production rates of $CO_2$ were measured by gas chromatography (Shimadzu GC-2014). The resulting depth-resolved rates are shown in Supplementary Fig. 3. The overall anaerobic rates were calculated by integrating over the entire anaerobic zone (depth = 0.5–6 cm). These rate estimates (11.6 and 7.9 μmol $d^{-1}$ for coarse and fine texture treatments, respectively) are in very good agreement with estimates based on the sum of individual anaerobic C oxidation pathways (10 and 8 μmol d$^{-1}$).

**Aerobic respiration.** The contribution of aerobic respiration to the overall C oxidation was calculated based on the difference between the total C oxidation rates and anaerobic respiration rates.

**Denitrification.** Denitrification rates were quantified by adapting an isotope-pairing technique specifically developed for flow reactors[37]. Here, $^{15}N$-labeled $NO_3^-$ was added, and the production of $^{15}N_2$ in the effluent was quantified. The rate of $^{15}N_2$ production was used to calculate the rate of denitrification based on the stoichiometry of the reaction given in Supplementary Table 3. For this purpose, a separate set of flow reactors were run for 30 days. On the 31st day, 10 μM of 99 at % $Na-^{15}NO_3^-$ was added to the influent solution, and $^{15}N_2$ in the effluent samples was monitored for the subsequent 8 days. The outflowing water was collected in a pre-evacuated 75-mL flask refilled with He, which was then heated to about 75 °C to strip $N_2$ from the water[38]. Gas samples for the determination of the isotopic composition of $N_2$ in the headspace were extracted with a high-pressure liquid chromatography syringe through a butyl rubber stopper, and stored in pre-evacuated exetainers flushed with He. Gas samples containing $^{15}N-N_2$ originating from denitrification were analyzed for their content of $^{29}N_2$ and $^{30}N_2$ isotopes by isotope ratio mass spectrometry (UC Davis, Stable Isotope Laboratory). Note that this approach only measured denitrification from $^{15}NO_3^-$, and denitrification from $^{14}NO_3^-$ (coupling of nitrification–denitrification of native $^{14}N$) is unaccounted for. However, the rates obtained by this method (0.93 and 0.78 μmol C d$^{-1}$ for coarse and fine texture treatments, respectively) are in very good agreement with C oxidation rates calculated based on the anaerobic soil volume in which denitrification occurred. Within the denitrifying zone (0.5–1 cm), C oxidation rates were calculated to be 1.1 and 0.7 μmol C d$^{-1}$ for coarse and fine texture treatments, respectively.

**Mn reduction rates.** The amount of Mn reduction over the course of the experimental period was calculated based on the concentration of HCl-extractable

$Mn^{2+}$ in the soil volume below aerobic and denitrifying zones (depth >1 cm) relative to the concentration measured at the start of the experiment ($t_0$). Mn was extracted by transferring 0.5 g (dry wt. equivalent) of soil to 15 mL centrifuge tubes in the anaerobic glove bag, adding 10 mL of 1 M HCl, shaking the sample for 1 h, and centrifuging at 4000×$g$ for 15 min. The supernatant was then filtered through 0.22-µm syringe filters, and Mn concentrations were determined using ICP-OES. HCl-extractable Mn was considered to be bivalent Mn. Following the stoichiometric relationships described in Supplementary Table 3, it was assumed that the reduction of 2 moles of $Mn^{2+}$ contributed to the production of 1 mole of $CO_2$.

**Iron reduction rates.** The amount of Fe reduction over the course of the experimental period was calculated based on the concentration of HCl-extractable $Fe^{2+}$ in the soil volume below aerobic and denitrifying zones (depth >1 cm) relative to the concentration measured at the start of the experiment ($t_0$). The total $Fe^{2+}$ concentrations were determined in the HCl extracts (described above) using a phenanthroline assay. According to the stoichiometric relationships described in Supplementary Table 3, it was assumed that the reduction of 4 moles of $Fe^{2+}$ contributed to the production of 1 mole of $CO_2$.

**Sulfate reduction rates.** Solid-phase analysis of the soils showed negligible total S concentrations (<7 µg g$^{-1}$ soil), $H_2S$ profiles across the full soil volume recorded using microsensors (Unisense, Denmark) showed no evidence of $H_2S$ production, and S XANES spectra showed no significant accumulation of FeS or S° species. We therefore concluded that sulfate reduction was negligible in our system.

**Methanogenesis.** The rates of $CH_4$ production were estimated based on anaerobic incubations (described above) of individual soil layers harvested from the soil volume below aerobic and denitrifying zones (depth >1 cm). The production rates of $CO_2$ were measured by gas chromatography (Shimadzu GC-2014). The resulting depth-resolved rates are shown in Supplementary Fig. 7. The overall anaerobic rates were calculated by integrating over the entire depth profile (1–6 cm). Based on the stoichiometric relationships described in Supplementary Table 3, we assumed that the production of 1 mole of $CH_4$ contributed 1 mole of $CO_2$.

**Thermodynamic constraints on carbon oxidation.** Anaerobic respiration is the final step in anaerobic organic matter decomposition. Anaerobic respiration follows depolymerization and fermentation reactions that produce organic compounds small enough to be assimilated into the cell. In anaerobic respiration, organic compounds of reactions are oxidized to $CO_2$ (i.e., mineralized) using alternative terminal electron acceptors such as $NO_3^-$, Mn(III,IV), Fe(III), and $SO_4^{2-}$. To evaluate the bioenergetics of this redox reaction, both electron donor and acceptor half-reactions have to be considered. Recently, a new formalism has been put forward that better captures the thermodynamic influence on C oxidation by explicitly considering the bioenergetics of the donor half-reaction and efficiency of electron transport chains[22,23]. As derived by Jin and Bethke[23], mineralization (or oxidation) rate can be expressed as

$$\text{Rate}_{\min} = R_{\max} X F_K F_T \qquad (2)$$

where $R_{\min}$ and $X$ are the maximum reaction rate and microbial biomass, respectively. The functions $F_K$ and $F_T$ are nondimensional and vary between 0 and 1. $F_K$ represents a microbe's ability to acquire and process the reactants, thus accounting for enzyme kinetics expressed within the typical Michaelis–Menten or Monod equations; it also encompasses mineral protection and physical isolation. What has not been considered is the catabolic energy yield, $F_T$, that links the rate of reaction to the thermodynamic driving force. The function $F_T$ represents a fundamental bioenergetic control on the microbial mineralization of organic compounds, and is dependent on the Gibbs free-energy term, $\Delta G_{rxn}$, for the overall reaction. $\Delta G_{rxn}$ is dependent on the reduction half-reaction of the terminal electron acceptor and the oxidation half-reaction for the carbon substrate.

LaRowe and van Cappellen[17] recently gave a formalism relating $\Delta G_{rxn}$ for the oxidation of a given organic compound to its NOSC. The NOSC for an organic compound can be derived from its oxidation half-reaction using the following relationship:

$$\text{NOSC} = -((-Z + 4a + b - 3c - 2d + 5e - 2f)/a) + 4 \qquad (3)$$

Here, $Z$ corresponds to the net charge of the organic compound, and the coefficients $a$, $b$, $c$, $d$, $e$, and $f$ refer to the stoichiometry of C, H, N, O, P, and S. We can thus predict $\Delta G_{rxn}$ for the oxidation of an organic compound simply through its composition. This concept provides an easy means to calculate the thermodynamic driving force, $F_T$, for the microbial oxidation of an organic compound when coupled to predominant electron acceptors in environmental systems. When coupled to oxygen, $F_T$ is close to 1, and the reaction is expected to proceed almost uninhibited for compounds spanning the full range of NOSCs. Consequently, the mineralization rate ($R_{\min}$) would be the same for reduced compounds with low NOSC (e.g., lipids) and more oxidized compounds with high NOSC (e.g., sugars) (Supplementary Fig. 8). However, when oxidation of these compounds is coupled to the reduction of Fe(III), the most abundant terminal electron acceptor in upland soil systems, a very different scenario results. $F_T$ for

sugars with NOSCs of around zero decreases to 0.70, reducing $R_{\min}$ by 30% relative to aerobic conditions. For reduced substrates such as lipids, this decrease is even more dramatic. $F_T$ for lipids with NOSCs of less than −1.7 is zero, and microbial oxidation is thermodynamically inhibited. As a consequence, microbial mineralization of abundant compound classes such as lipids and fatty acids may be thermodynamically limited if not completely inhibited under anaerobic conditions[24].

**Carbon characterization.** To isolate carbon pools of increasing bioavailability, a fractionation procedure was employed that relies on a density gradient established in sodium polytungstate (SPT) solution. The total C in bulk samples and density isolates was determined using an elemental analyzer (NA1500, Carlo Erba), while DOC concentrations in the pore water were determined on a Shimadzu TOC-L. Further details on the density fractionation procedure and the molecular characterization of solid and dissolved organic carbon forms can be found in the respective sections below.

**Density fractionation of reactor and field samples.** To isolate SOC pools of increasing bioavailability, a fractionation procedure was employed that relies on a density gradient established in SPT solution. Our fractionation method separated soil from each zone into two fractions: (1) a light fraction consisting primarily of particulate organic carbon (POC) and (2) a heavy fraction, comprising mineral-associated organic carbon (MAOC). We followed the procedure of Moni et al.[39] with modifications as described here. POC was separated from 3 g of bulk soil by floatation in 12 mL of a low-C and -N sodium polytungstate (SPT-0, Geoliquids Inc., Prospect Hills, IL) with a density of 1.65 g cm$^{-3}$. The heavy fraction was separated from the residual soil using 12 mL of SPT solution with a density of >1.65 g cm$^{-3}$ that was collected as the remaining soil pellet. Reactor samples were weighed, fractionated, and filtered in an anaerobic glove bag directly after the reactor harvest to preserve the redox state and avoid precipitation reactions triggered by the presence of oxygen. Soils were mixed with anoxic SPT solution and centrifuged in airtight tubes at 4000×$g$ for 60 min. POC was collected on 0.8-µm nylon filters (Whatman nylon filter membranes, 47 mm) and then rinsed with distilled, deionized, and anoxic $H_2O$ to remove residual SPT. This process was repeated two times to isolate residual POC. The remaining pellet (i.e., MAOC) was washed repeatedly with distilled, deionized anoxic $H_2O$ in centrifuge tubes (three cycles with 10 mL of $H_2O$, centrifuged at 4000×$g$ for 15 min). Both fractions were flash frozen in liquid $N_2$ and lyophilized for further analysis. For the reactor experiment, samples collected from the aerated zones (depth = 0–0.5 cm) and the most reduced zones (4–6 cm) were fractionated. In principle, field samples were fractionated by the same procedure. However, in this case, 12 g of soil was suspended in 40 ml of SPT and three replicate fractionations were carried out.

**Carbon NEXAFS analyses.** For C NEXAFS analysis of bulk samples and density isolates, subsamples were gently ground, slurried in $H_2O$, deposited onto pre-cleaned In foils, and air-dried at room temperature. Pore water samples were directly deposited onto In foils and dried. C NEXAFS spectra were collected using the spherical grating monochromator (SGM) beamline 11ID−1 at the Canadian Light Source (CLS, Saskatoon, Saskatchewan, Canada)[40]. In order to minimize X-ray exposure, the spectra were collected in step scan mode (in 0.25-eV steps from 270 to 320 eV) with a dwell time of 20 ms. After each scan, the beam was moved to a new spot on the sample collecting a total of 50–70 scans for each sample. The beamline exit slit was set at 25 mm, and fluorescence yield data were collected using a two-stage microchannel plate detector. After averaging the scans for each sample, the pre-edge region (270–275 eV) of the average spectrum was set to zero (baseline normalization), and the resulting spectrum ($I$) was normalized to the beamline photon flux ($I_o$) recorded for a separate Au reference foil. The spectrum was referenced to the carboxylic acid peak (288.5 eV) of a citric acid standard for energy calibration. Edge step normalization was performed in Athena[41] using a pre-edge region of 270–278 eV, a post-edge region of 310–320 eV, and an $E_0$ value of 290 eV.

The relative abundance of functional groups, specifically that of carboxyl and aromatic C, was determined by peak deconvolution using Athena's peak fit function[41]. Peak positions were assigned according to conventions reported by Schumacher et al.[42] and Solomon et al.[43]. Positions of Gaussian peaks, their full-width at half-maximum, and the arc tangent function were fixed. However, peak magnitude was allowed to vary freely during the fitting process. Parameters were iteratively adjusted until optimal fits were achieved for all spectra in the data set. The final parameter set is shown in Supplementary Table 4. All spectra were fitted with this final parameter set (Supplementary Table 4), and the results are summarized in Supplementary Table 5 and Supplementary Fig. 9.

**Recovering water-extractable organic carbon.** Water-extractable organic carbon as a proxy for bioavailable compounds was obtained for spectroscopic and mass spectrometric examination by mixing 20 g of the field soils with 45 mL of Milli-Q water. The suspension was vortexed for 30 s, agitated at 100 rpm for 1 h in the dark, and centrifuged at 4000×$g$ for 60 min. The supernatant was subsequently filtered through 0.2-µm nylon membranes. Extracted organic carbon was quantified using a TOC analyzer (Shimadzu TOC-L). A first set of subsamples was flash frozen and submitted for mass spectrometry analysis at the Environmental Molecular Science

Laboratory (EMSL) at Pacific Northwest National Lab. A second subset was flash frozen and lyophilized in the dark for NMR spectroscopy analysis.

**FT-ICR-MS analysis of water extracts**. Frozen WEOC samples were submitted to the Environmental Molecular Science Laboratory at Pacific Northwest National Lab for FT-ICR-MS analysis. A 12-T Bruker SolariX FTICR mass spectrometer was used to collect high-mass resolution and mass accuracy spectra of the organic molecules in the WEOC. A standard Bruker ESI source was used to generate negatively charged molecular ions. Samples were introduced to the ESI source equipped with a fused silica tube (200 μm i.d.) through a syringe pump at a flow rate of 3.0 μL min$^{-1}$. The experimental conditions were as follows: needle voltage, +4.4 kV; Q1 set to 150 m/z; and the heated resistively coated glass capillary operated at 180 °C. In total, 96 individual scans were averaged for each sample and internally calibrated using OM homologous series separated by 14 Da (–CH$_2$ groups). The mass measurement accuracy was typically within 1 ppm for singly charged ions across a broad m/z range (i.e., 200 < m/z < 1200). Chemical formulas were assigned using the Compound Identification Algorithm (CIA) as described by Tfaily et al.[44].

Using the assigned formulae and the stoichiometric information contained therein, we calculated the NOSC following a simplified version of Eq. 3:

$$NOSC = -((-Z + 4C + H - 3N - 2O + 5P - 2S)/C) + 4 \qquad (4)$$

where $Z$ corresponds to the net charge of the organic compound here assumed to be zero, and $C$, $H$, $N$, $O$, $P$, and $S$ refer to the stoichiometric numbers of each respective element provided by FT-ICR-MS analysis. This approach allows us to examine whether compounds with low NOSC values become preferentially preserved in anaerobic microsites prevalent in fine-textured upland soils. NOSC values were calculated for each mass peak individually. Figure 6 shows mean, median, one standard deviation, and 95% confidence intervals of NOSC values of all identified masses for each sample.

**Solution-state $^{13}$C NMR spectroscopy of water extracts**. We used solution-state $^{13}$C NMR spectroscopy, with proton decoupling during acquisition, to analyze the functional group composition of WEOC in both soils. Lyophilized WEOC samples were dissolved in 600 μL of 99.9% D$_2$O with 2.5 mM d6-DSS. Data were collected on an Agilent VNMRs 750-MHz (proton frequency) spectrometer at EMSL using a Broad Band X-Observe probe tuned to $^{13}$C (188.2 MHz). The NMR experimental parameters used were a 45° pulse (5.5 μs), 1.0-s acquisition time, a spectral window of 37,878.8 Hz, a 5-s recycle delay, 25 °C, and 12,800 scans. Waltz16 decoupling on $^1$H was applied during the acquisition. The total experimental acquisition time for each experiment was ~21 h. The spectra were referenced to d6-DSS at 0 ppm. Data were processed in MestRenova, zero filled to twice the final size, and apodized with 10 Hz of line broadening. The relative abundances of functional groups (Supplementary Fig. 4) were quantified by integrating across ppm regions, as defined by Clemente et al.[19].

**Data availability**. The authors declare that the data supporting the findings of this study are available within the article, and its Supplementary Information file, and from the corresponding author on reasonable request.

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

## Acknowledgements

We are grateful to the analytical assistance from G.C. Li and D. Turner. Synchrotron spectroscopy reported in this paper was performed at the Canadian Light Source (CLS), which is supported by the Canada Foundation for Innovation, Natural Sciences and Engineering Research Council of Canada, the University of Saskatchewan, the Government of Saskatchewan, Western Economic Diversification Canada, the National Research Council Canada, and the Canadian Institutes of Health Research. The authors thank J. Dynes and T. Regier for their assistance at CLS's SGM beamline. A portion of this research was performed using EMSL, a DOE Office of Science User Facility sponsored by the Office of Biological and Environmental Research and located at Pacific Northwest National Laboratory. We thank EMSL staff members M. Tfaily and P. Reardon for FT-ICR-MS and NMR analyses, respectively. This work was supported by the US Department of Energy, Office of Biological and Environmental Research, Terrestrial Ecosystem Program (Award Number DE-FG02-13ER65542), and Subsurface Bio-geochemistry Program (Award Number DE-SC0016544).

## Author contributions

This work was originally conceived by S.F. and M.Kei. with additional input from M.Kle. and P.N. Laboratory experiments were conducted by M.Kei.; soil sampling and characterization were carried out by T.W. and M.Kle. M.Kei. performed chemical analyses on laboratory and field samples, with input from P.N. The manuscript and supporting information were written by M.Kei. with input from all coauthors, particularly S.F.

## Additional information

**Competing interests:** The authors declare no competing financial interests.

