## [Peer Review File · Nature Communications]

Reviewers' Comments:

Reviewer #1:

Remarks to the Author:

I really appreciated the detailed and thoughtful replies of the authors, addressing the comments and suggestions made by the reviewers. The manuscript has improved significantly in clarity. After some final polishing and editing I would recommend publication.

Details: In the supplementary material „bookmarks“ should be edited and some extra spaces removed.

Reviewer #2:

Remarks to the Author:

In the original review of this manuscript, I very much appreciated the very high quality controlled work and novel mechanistic insights it afforded into thermodynamic constraints leading to the protection of compounds under anaerobic conditions. I was less comfortable with the extrapolation of the results to carbon storage in upland soils and the authors have done a thorough and compelling job in revising this extrapolation to satisfy all three reviewers. In doing so, the real novelty of their work is now front and center. One reviewer questioned whether we already understood the thermodynamic constraints the authors reveal - I think it is now clear in the revised script that although we might know the pattern it generates, we did not have the understanding of this process explanation. By providing this process-level understanding I am confident in stating that the large majority of those who work on soil carbon dynamics will be engaged, interested and challenged by this script, given that it shows that the new emerging model of soil carbon stabilization emphasizing microbial-mineral interactions, is likely far from complete.

Reviewer #4:

Remarks to the Author:

It is well known that measurements of oxygen content along the soil profile are difficult to achieve and its role in the mineralization process is problematic to assess. So talking about anaerobic microsites is very speculative and not supported by a robust and well design experiments. I understood that the anaerobic microsite was not the primary objective of the paper. Thus why the authors are making such a big deal about phenomena that controls C sequestration/ preservation without quantitative scientific evidences of their existence in the soil. We have only assumption/supposition of their existence (cited review paper from the same authors of this paper). We need to know how much they represent (including their distribution: top vs. subsoil) along the soil profile.

In the absence of strong evidences I'm not convinced that the applied experiment design of this study (using disturbed soil) is strong enough to show the thermodynamic constraints of lipid protection or the C release. The process under anaerobic conditions related to C mineralization has been widely studied and they are well known. Moreover measurements carried out at the laboratory level are completely disconnected from the reality. What are the proofs of what is found in the laboratory is really matching the real world? We all know that all the chemical

characterization of preserved molecules described in the paper is well documented in the literature (e.g. studies on organo-mineral interactions, studies on redox chemistry...). So stating some very well documented and implemented processes is not professional (e.g. L 99-105; L 139...).

General comment

I realize that the authors have been successful in their style of writing in the past. However, there are several instances where the results would merely be a synthesis of material that is already published in the literature (e.g. a translation/adaptation/application of well known process on organo-mineral interactions or redox reactions studies). The whole story of the paper is all about a phenomena in which (i) the authors provide no quantitative scientific evidences of their existence in the soil and (ii) how they may controls C preservation and release.

RESPONSE TO REVIEWERS' COMMENTS

Reviewer #1:

Comment: I really appreciated the detailed and thoughtful replies of the authors, addressing the comments and suggestions made by the reviewers. The manuscript has improved significantly in clarity. After some final polishing and editing I would recommend publication.

Details: In the supplementary material „bookmarks“ should be edited and some extra spaces removed.

Authors' response: *The citations in the SI were corrected and the extra spaces removed.*

Reviewer #2:

Comment: In the original review of this manuscript, I very much appreciated the very high quality controlled work and novel mechanistic insights it afforded into thermodynamic constraints leading to the protection of compounds under anaerobic conditions. I was less comfortable with the extrapolation of the results to carbon storage in upland soils and the authors have done a thorough and compelling job in revising this extrapolation to satisfy all three reviewers. In doing so, the real novelty of their work is now front and center. One reviewer questioned whether we already understood the thermodynamic constraints the authors reveal - I think it is now clear in the revised script that although we might know the pattern it generates, we did not have the understanding of this process explanation. By providing this process-level understanding I am confident in stating that the large majority of those who work on soil carbon dynamics will be engaged, interested and challenged by this script, given that it shows that the new emerging model of soil carbon stabilization emphasizing microbial-mineral interactions, is likely far from complete.

Reviewer #4:

Comment: It is well known that measurements of oxygen content along the soil profile are difficult to achieve and its role in the mineralization process is problematic to assess. So talking about anaerobic microsites is very speculative and not supported by a robust and well design experiments. I understood that the anaerobic microsite was not the primary objective of the paper. Thus why the authors are making such a big deal about phenomena that controls C sequestration/ preservation without quantitative scientific evidences of their existence in the soil. We have only assumption/supposition of their existence (cited review paper from the same authors of this paper). We need to know how much they represent (including their distribution: top vs. subsoil) along the soil profile.

Authors' response: *We added a statement to convey that the lack of quantitative data documenting the abundance of anaerobic microsites should be addressed in future work.*

Changes Made: L232-234: *“To assess the potential magnitude of this effect, quantitative data documenting the abundance of anaerobic microsites and the amount of carbon protected therein is urgently needed.”*

Comment: In the absence of strong evidences I'm not convicted that the applied experiment

design of this study (using disturbed soil) is strong enough to show the thermodynamic constraints of lipid protection or the C release. The process under anaerobic conditions related to C mineralization has been widely studied and they are well known. Moreover measurements carried out at the laboratory level are completely disconnected from the reality. What are the proofs of what is found in the laboratory is really matching the real world? We all know that all the chemical characterization of preserved molecules described in the paper is well documented in the literature (e.g. studies on organo-mineral interactions, studies on redox chemistry...). So stating some very well documented and implemented processes is not professional (e.g. L 99-105; L 139...).

I realize that the authors have been successful in their style of writing in the past. However, there are several instances where the results would merely be a synthesis of material that is already published in the literature (e.g. a translation/adaptation/application of well known process on organo-mineral interactions or redox reactions studies). The whole story of the paper is all about a phenomena in which (i) the authors provide no quantitative scientific evidences of their existence in the soil and (ii) how they may controls C preservation and release.

Authors' response: We clearly have a strongly different opinion than the reviewer, and we believe the evidence in the literature supports our view and justifies our experimental approach—as noted by the other reviewers. We indicated that the precise contribution of anaerobic microsities to global soil carbon fluxes is outstanding and should be a future directive. There is, however, ample evidence for anaerobic microsite contributions within upland soils, ranging from electrode measurement of oxygen to preservation of reduced carbon compounds—as described within our manuscript. Our work provides several novel insights that, as Reviewer 2 indicates, underpin why carbon is preserved due to anaerobic conditions, what will happen upon disturbance, and the rate at which transformation will occur. Moreover, we marry laboratory experiments with field measurements. The laboratory experiments provide a quantitative framework for the metabolic impacts of anaerobic microsities and the thermodynamic mechanisms controlling carbon utilization; field measurements are then used to verify that the preservation mechanisms are operational. Our work is therefore not abstract, as the reviewer indicates, but grounded with field measurement and, we believe, will change the way microbial processing of soil carbon is viewed.